# Can Simple Tests Prior to Endoscopy Predict the OLGA Stage of Gastritis?

**DOI:** 10.3390/healthcare8030230

**Published:** 2020-07-24

**Authors:** Ertan Bulbuloglu, Hasan Dagmura, Emin Daldal, Alev Deresoy, Huseyin Bakir, Ugur Ozsoy, Ali Ihsan Saglam, Osman Demir

**Affiliations:** 1General Surgery Department, Faculty of Medicine, Kahramanmaras Sutcu Imam University, 46000 Kahramanmaras, Turkey; ertanbulbuloglu@ksu.edu.tr; 2General Surgery and Surgical Oncology Department, Dumlupinar University Kütahya Evliya Çelebi Training And Research Hospital, 60250 Tokat, Turkey; 3General Surgery Department, Gaziosmanpasa University, 60250 Tokat, Turkey; emin.daldal@gop.edu.tr; 4Pathology Department, Gaziosmanpasa University, 60250 Tokat, Turkey; alevderesoy@yahoo.com; 5General Surgery and Surgical Oncology Department, Gaziosmanpasa University, 60250 Tokat, Turkey; drhbakir@gmail.com; 6Resident of General Surgery Department, Gaziosmanpasa University, 60250 Tokat, Turkey; drugurozsoy@gmail.com (U.O.); saglam_ts@hotmail.com (A.I.S.); 7Biostatistics Department, Gaziosmanpasa University, 60250 Tokat, Turkey; mosmandemir@hotmail.com

**Keywords:** Operative Link for Gastritis Assessment, gastric cancer, endoscopy, pain, numerical rating scale, neutrophil to lymphocyte ratio

## Abstract

Gastritis is a progressive disease that evolves from a non-atrophic to atrophic state and progresses through intestinal metaplasia, with some cases leading eventually to gastric cancer. Since gastritis by definition is an inflammatory process of the mucosal lining of the stomach and is usually associated with pain, we aimed to identify any association between the severity of gastritis and pain and a simple inflammatory marker derived from a complete blood count (CBC). This was a prospective cross-sectional study which enrolled 155 consecutive adult patients who underwent an upper endoscopy. Prior to the endoscopy, all patients were given a questionnaire, numerical rating scale (NRS) and complete blood count evaluation. The biopsy was obtained from the gastric mucosa according to the modified Sydney classification and scored with the Operative Link for Gastritis Assessment (OLGA) scoring system. The results showed a significant correlation between NRS and intestinal metaplasia (*p* < 0.01); moreover, a correlation was also found between the NRS and OLGA stage (r = 0.469, *p* < 0.001). A nonlinear curve was constructed for OLGA stage estimation according to NRS scores (r^2^ was found to be 0.442, with *p* < 0.001). The results also showed a correlation between the neutrophil to the lymphocyte ratio (NLR) and OLGA stage (*p* < 0.01). No correlation was found between the other gastric parameters and NLR (*p* > 0.05). *Helicobacter pylori* positivity did not correlate with NRS and NLR. As a conclusion, pain measured by NRS and NLR, which are simply calculated from the CBC prior to endoscopy, may be used to predict OLGA stages and estimate the severity of gastritis in endoscopy patients.

## 1. Introduction

Gastritis is the injury to the mucosal lining of the stomach, which is manifested in the form of an inflammatory process accompanied by damage to the mucus-lined barrier that protects the mucosal wall. Even though the cause of gastritis is undetermined in most cases, the only well-known etiology is infection caused by *H. pylori* [1]. Other causes that may be listed are drugs such as NSAID, excessive alcohol use, auto-immune diseases, parasitic infestations, viral infections, granulomatous diseases such as Crohn’s disease, sarcoidosis, stress (trauma, burns, major injury) and malnutrition. During upper endoscopy, findings of gastritis may include erythema, mucosal erosions, the absence of rugal folds and the presence of visible vessels; however, none of these features are diagnostic for gastritis. Other than the classical classification of gastritis into acute and chronic, further classifications have been proposed: OLGA (Operative Link for Gastritis Assessment) and OLGIM (Operative Link on Gastric Intestinal Metaplasia Assessment) are exclusively used for identifying and assessing patients with chronic atrophic gastritis and intestinal metaplasia, respectively, who are at higher risk of developing gastric cancer [1,2,3,4]. The degrees of atrophy and metaplasia are directly proportional to the risk of developing gastric cancer.

Pain is described as a highly unpleasant physical sensation that may be caused by illness or injury. In the case of gastritis, pain is the most common symptom and the initial clinical manifestation. Since gastritis, by definition, is an inflammatory process of the mucosal layer of the stomach, we expect a reflection of this process in the circulating inflammatory parameters in the blood stream. Epigastric pain, which may be accompanied by mild signs of inflammation, is the initial and possibly the only symptom of gastritis. To the best of our knowledge, no prior study has been conducted which has focused on the possible association between pain as a subjective tool and histopathological findings in patients with gastritis.

Currently, the only way to measure patients’ pain is by subjective assessment; this subjectivity causes a problem, because patients may rank the same pain differently, thus producing different results. Good communication between the patient and the investigator may decrease the subjectivity issue. Gender, race and the state of consciousness may affect the outcome of pain assessment. In the current study, a standard questionnaire for the evaluation of pain intensity was used in order to decrease any sort of possible bias.

The neutrophil to lymphocyte ratio (NLR) has been reported to be a prognostic factor of subclinical inflammation in various disease states. This emerging trend of using NLR as a predictor or diagnostic tool for the severity of an illness had led to its widespread use in a wide spectrum of diseases, namely as a marker of systemic inflammation, such as in cancer treatment or coronary artery bypass grafting [5,6,7,8].

The aim of this study is to reveal any correlation between pre-procedural simple measurements (pain intensity and NLR) and the histological severity of gastritis.

## 2. Methods and Materials

This study was designed as a cross-sectional prospective study and enrolled 200 consecutive adult patients who underwent upper endoscopy. Exclusion criteria included previous surgery, malignancy, gastrointestinal bleeding, emergency cases, foreign body ingestion, a history of peptic ulcers, presence of active and chronic infection and presence of a hematological disease.

Out of 200 patients, 45 were excluded from the study due to several reasons, namely inadequate sampling (insufficient quantity or quality of the taken specimen) or intolerance of the procedure by the patient (Figure 1). This study was conducted between December 2018 and September 2019 in the endoscopy unit of the University Affiliated Teaching Hospital

The Ethical Committee of the University Medical Faculty approved the study (Ethical Committee number 18-KAEK-166). Written informed consent for endoscopic procedures was obtained before the examination from all patients.

The endoscopy unit consists of gastroscopy, colonoscopy, ERCP and endoscopic ultrasound; for the gastrointestinal section, the annual average number of upper endoscopies and colonoscopies performed is more than 2800 procedures per year. The staff consisted of gastroenterologists, general surgeons and surgical oncologists assisted by specialized endoscopy nurses. To reduce the bias of endoscopic evaluation, the study patients underwent endoscopy by two certified endoscopists (HD, ED) under the supervision of a senior member of staff (IO). All endoscopic findings were noted on a pre-formed data sheet. A biopsy was obtained from the gastric mucosa according to the modified Sydney classification [9]. In brief, this proceeded as follows: from the antrum, two biopsies were performed and noted as “A1 and A2”; from the incisura angularis, a single biopsy was performed, noted as “A3”; and one biopsy each from the lesser curvature and the greater curvature was performed, noted as “C1” and “C2”, respectively. The collected biopsy specimens were fixed immediately in 10% buffered formalin and transferred to the pathology unit on the same day. The biopsies were reviewed by only one gastrointestinal pathologist (AA) and assessed according to OLGA and the modified Sydney classification.

### 2.1. Sedation

All patients undergoing endoscopic examinations required a fasting period of a minimum of 6 h. The endoscopy was routinely performed under sedation with midazolam and pethidine in varying doses to obtain moderate sedation levels during the procedure. Midazolam was given at a dose of 0.03 mg/kg and pethidine at a dose of 1.0 mg/kg for 60 s intravenously. During endoscopies, additional midazolam and pethidine doses were administrated when indicated depending on patient discomfort and vital signs, at up to 6 mg and 100 mg, respectively. Reversal agents (flumazenil, naloxone) were used in the case of sedation-related complications. All sedation procedures were executed by a nurse practitioner under the guidance of the endoscopist.

### 2.2. Monitoring

Patients were given supplemental oxygen (4 L/min) through a nasal catheter to keep the saturation over 90% if necessary. The pulse rate, arterial blood pressure, oxygen saturation and level of consciousness were monitored by the nurse practitioner.

### 2.3. Endoscopy Procedure

All endoscopic procedures were performed by experienced endoscopists. All endoscopists used the same type of endoscope (OLYMPUS). Photographs of the esophagus, stomach and duodenum segments were taken and archived for documentation and assessment.

### 2.4. Sample Size Calculation:

For the numerical rating scale (NRS) single sample level, 156 people were included in this work with 80% power, a 5% margin of error and 0.1 effect size. Sample size was calculated with the G * power (version 3.1.2) program.

For the NLR simple sample level, 156 people were included in this work with 80% power, a 5% margin of error and 0.2 effect size. Sample size was calculated with the G * power (version 3.1.2) program.

A total of 200 patients were included in the study; we predicted that the total patient population loss would be around 25%.

### 2.5. Data Collection

Before the examination, all patients completed a questionnaire, which consisted of the following:

A—Demographic data and other characteristics: age, sex, file number, main complaint, any previous peptic ulcer treatment, drugs, chronic disease and surgical abdominal history.

B—Pain assessment questionnaire:NRS;NRS end points (0–10);Frequency of administration of the test;Time period to be rated;Type of pain intensity rated by the participant (average, least, worst and current);Pain location in another part of the body.

The calculation of the NLR (neutrophil to lymphocyte ratio) was as follows:

NLR = absolute number of neutrophils/absolute number of lymphocytes

### 2.6. Statistical Methods

The statistical analysis of the data obtained from the study was performed using SPSS (Version 22.0, SPSS Inc., Chicago, IL, USA). Descriptive statistics were presented with mean ± standard deviation and median (min–max) according to the data distribution for continuous variables. Descriptive statistics of categorical variables were reported as numbers and percentages (%). The distribution of the normality of data for statistical test selection was evaluated by the Shapiro—Wilk test. Comparisons of two independent groups were performed by the Mann—Whitney U test since the data were not distributed normally. The correlations between continuous variables were evaluated with the Spearman correlation coefficient according to the data distribution. For the modeling between the NRS and OLGA stage, the nonlinear curve (quadratic) was fitted. A Chi-square test was used to investigate the relationship between categorical variables and to compare proportions. The level of statistical significance was accepted as *p* < 0.05.

## 3. Results

A total of 155 patients were included in the study. The mean age of the patients was 49.1 ± 15.2 (min–max: 18–81) with 54.2% females (*n*: 84).

The results of the correlation analysis between NRS and gastric parameters are presented in Table 1. There was a significant weak correlation between NRS and subgroup scores as well as intestinal metaplasia (*p* < 0.01; Table 1). A weak—moderate correlation was also found between NRS and OLGA stage (r = 0.469, *p* < 0.001; Table 1). A nonlinear curve was constructed for OLGA stage estimation according to NRS scores (Figure 2). According to the obtained equation, a formulation if OLGA = 1.907 − 0.434 ∗ NRS + 0.065 ∗ NRS^2^ was obtained between NRS and OLGA (r^2^ = 0.442, *p* < 0.001).

There was no correlation between other gastric parameters (intestinal metaplasia, inflammation, activity) and NRS (*p* > 0.05).

There was a statistically significant weak correlation between the NLR and OLGA stage (*p* < 0.01; Table 2). No correlation was found between the other gastric parameters and NLR (*p* > 0.05). The details were presented in Table 2.

The comparison of Helicobacter Pylori (HP) positivity rates according to NRS, NLR and NRS/NLR groups is given in Table 3. HP distributions were statistically similar in the groups (*p* = 0.354, *p* = 0.156, *p* = 0.194).

The comparison of gastric parameters according to NLR and NSR groups (high, low) is shown in Table 4. The C score and OLGA stage were found to be significantly different between the groups (*p* = 0.001, *p* = 0.007, Table 4). C scores and OLGA stages were significantly higher in the positive group; other scores were similar between the groups (*p* > 0.05, Figure 3). Demographic and clinical descriptive statistics of patients (Table 5).

## 4. Discussion

The purpose of this study was to detect gastric lesions, such as atrophy and metaplasia, in patients with gastritis by using simple measurements such as NRS and the complete blood count. Are these two parameters sufficient to orient the physician to make a decision whether to order a relatively “urgent” upper endoscopy for histopathological verification or to go forward with medical treatment? Our study showed that there was a correlation between NRS and intestinal metaplasia. Moreover, a correlation was also found between the NRS and OLGA stage. Similarly, the results of the correlation analysis between NLR and gastric parameters showed a statistically significant weak correlation between the NLR and OLGA stage. In this prospective study, we have found that there is a relationship between NRS and the level of gastric atrophy as well as metaplasia; this correlation is in the a nonlinear form, which implies that NRS can predict the level of atrophy—i.e., the OLGA stage—by using a simple equation.

Gastritis is a widely used general term to denote the inflammation of the mucosal lining of the stomach. Although various factors are listed in its etiology, the most common and well-described one is *Helicobacter pylori* colonization. As the inflammation progresses, the damage to the epithelial lining of the stomach becomes severe. The stages of inflammation were proposed by Correa as chronic atrophic gastritis, intestinal metaplasia, dysplasia and eventually malignancy [10]. Chronic atrophic gastritis is characterized by the loss of cells including G and parietal cells. A further progression of the inflammatory process would lead to the replacement of gastric epithelial cells by the intestinal type of cells, which is known as intestinal metaplasia. Pain is one of the cardinal signs of inflammation [11]. The induction of inflammation could evoke a pain stimulus through the nerve fibers in injured mucosa [12]. Stomach pain is a visceral type of pain originating from the mucosal surface and carried by afferent fibers which are mainly unmyelinated C-fibers. They traverse through the spinal cord, resulting in a dull, poorly localized pain. In order to assess the pain intensity, various tools have been described; NRS is a widely used subjective tool that can be used for pain assessment in this kind of situation. In the current study, NRS was used as a tool to measure pain intensity; an NRS value greater than 4 on a scale of 11 (0–10) was considered to be a positive result [13]. In the adult population, an OLGA score higher than two is linked to an increased risk of gastric cancer, especially when associated with metaplasia [14]. In this study, the NRS test result was found to be directly proportional to the degree of atrophy, whether regarding an isolated condition at the level of the antrum (A) or corpus (C) or disseminated to involve all the stomach; this finding provides physicians with an idea about the degree of severity of gastric atrophy. Atrophic gastritis with an OLGA score above two implies a multifocal atrophy at the mucus-secreting and/or oxyntic level or a diffuse disease state involving both the mucus-secreting site and the oxyntic region. In this condition, a significantly higher risk of developing gastric cancer (GC) has been reported; thus, endoscopic surveillance programs should consequently concentrate on OLGA stage three and four patients [15,16]

Similarly, the intensity of the inflammatory process can be measured by using parameters derived from the complete blood cell count. The most commonly used derivative is NLR. The possible inflammatory markers that can be used to evaluate gastritis are the interleukin family—mainly IL-6—and other proinflammatory cytokines, namely the tumor necrosis factor alpha (TNF-α), IL-1, and IL-8, C-reactive protein (CRP), platelets and neutrophils. The neutrophil to lymphocyte ratio (NLR) is a parameter that was developed to assess the inflammatory condition of a subject. Recently, it has proven to be a useful marker in predicting, assessing and stratifying several diseases and conditions such as inflammatory diseases and benign or malignant maladies, as well as in the prediction of postoperative mortality and complications [17,18,19,20]. Our findings suggested that there is a correlation between NLR and atrophy as well as metaplasia. NLR values vary according to age, associated chronic diseases, habits and many other factors. The cutoff value in this study was set as 3.5 according to the study conducted by Forget et al. Ref. [21] identified the normal NLR value in the non-geriatric, otherwise healthy adult population, to be between 0.78 and 3.53. Similarly, a correlation exists between NLR and gastric atrophy; however, no correlation was found between NLR and metaplasia. Atrophy can be associated with metaplasia, as it can be without metaplasia [1,22], and thus we do not expect metaplasia in every case of atrophy; in this study, concurrent atrophy and metaplasia was found in 21.9% of cases. However, a study conducted by Bellolio et al. found that 99% of patients with atrophy also had metaplasia [23]; this can be explained by the etiology of gastritis, ethnic factors and environmental factors.

In this study, we did not find any correlation between the intensity of gastric pain and *H. Pylori* colonization; this finding is consistent with many other studies conducted in the pediatric population, including those of Reifen et al. and Fiedorek et al. who evaluated recurrent epigastric pain and *H. Pylori* infection by endoscopic or urea test verification [24,25].

Few patients simultaneously had both high NLR and high NRS values. These patients with both NLR and NRS high value tests were associated with a statistically significant elevated OLGA score with a mean = 2.14, *p* < 0.007 (Table 4), indicating that these patients had a tendency to have a higher degree of atrophy than those with low-value tests; this may have an impact in predicting patients with premalignant lesions, who are at greater risk of gastric cancer.

Additionally, a recent paper showed that the combination of the NLR and platelet to lymphocyte ratio might even be a better predictor of HP infection-related gastrointestinal complications than NLR alone [26].

The limitations of our study may include the following factors: first, this work is a single-center study in a tertiary university hospital; second, the NRS test, which is a subjective in nature, was performed just before the upper endoscopic examination and may be responsible for an implicit bias that may vary dramatically; third, there is a possible inter-observer variability associated with OLGA assessment, and this was performed by a single pathologist; finally our paper was limited by the small sample size. Some of these limitations may be overcome by increasing the sample size for the study, performing a multi-centric rather than a single-center study and increasing the number of pathologists to eliminate the inter-observer variability factor.

The shortage of endoscopy appointments, which means a prolonged endoscopy waiting time, is a major public health issue. Due to the steadily growing demand for endoscopic examinations compared to the number of procedures performed, there is an increasing waiting time for endoscopic procedures; in a study, conducted by Harewood et al. in a major teaching hospital, this time was estimated to be 40 weeks [27]. By using these two simple tests, one can partially contribute to solving this public health problem by reducing the load by omitting unnecessary endoscopic examinations.

## 5. Conclusions

In patients with gastritis, NLR as well as NRS tests can be used as simple clinical methods to predict the OLGA stage; however, we should keep in mind that the strength of correlations found in this study are weak–moderate. By means of these two non-invasive and low-cost tests, one could identify high-risk patients, promoting them to “early” endoscopic investigations for histopathologic verification, especially for those with a high NLR combined with a high NRS value.

## Figures and Tables

**Figure 1 healthcare-08-00230-f001:**
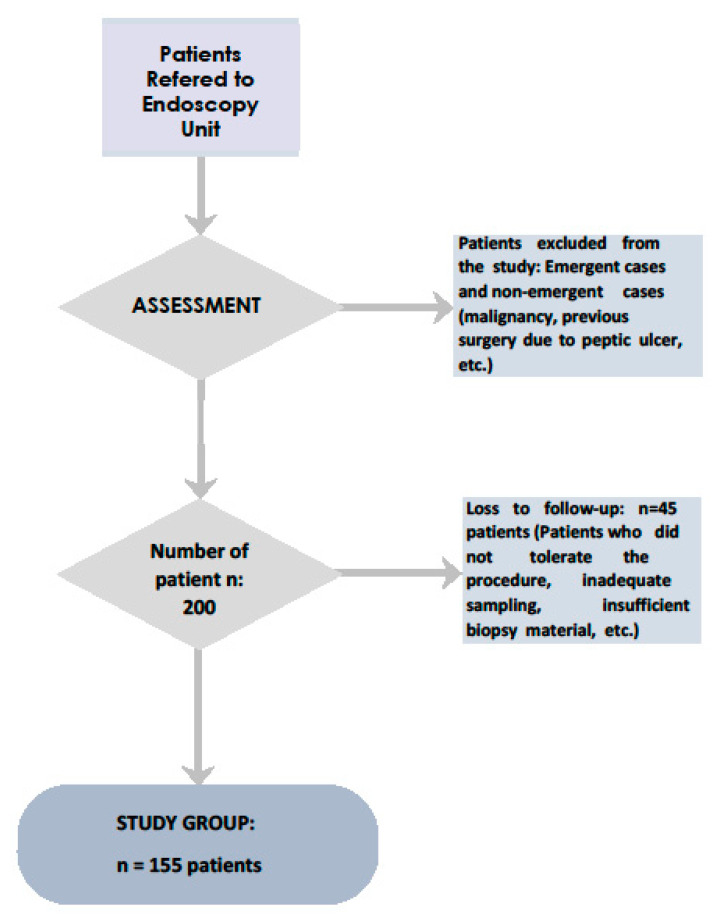
Flowchart of the selection of the study group patients.

**Figure 2 healthcare-08-00230-f002:**
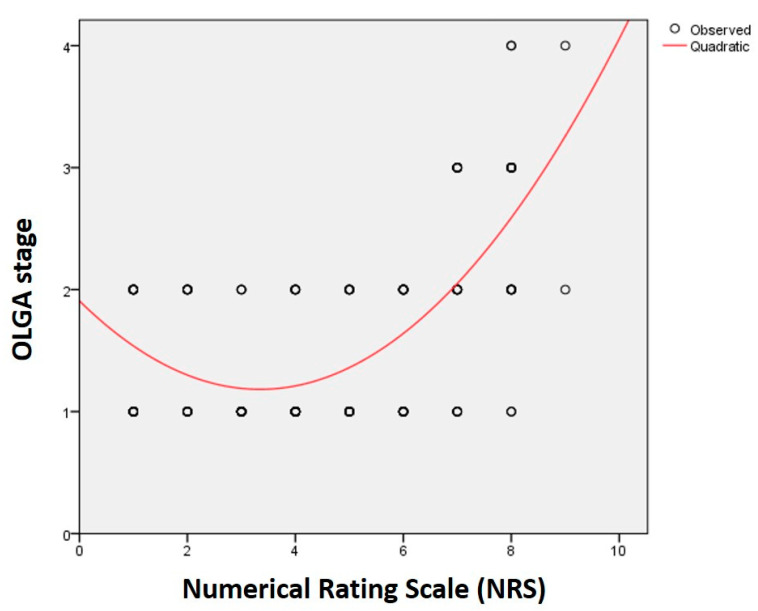
Curve estimations between numerical rating scale (NRS) and Operative Link for Gastritis Assessment (OLGA) stage (OLGA = 1.907 − 0.434 ∗ NRS + 0.065 ∗ NRS^2^).

**Figure 3 healthcare-08-00230-f003:**
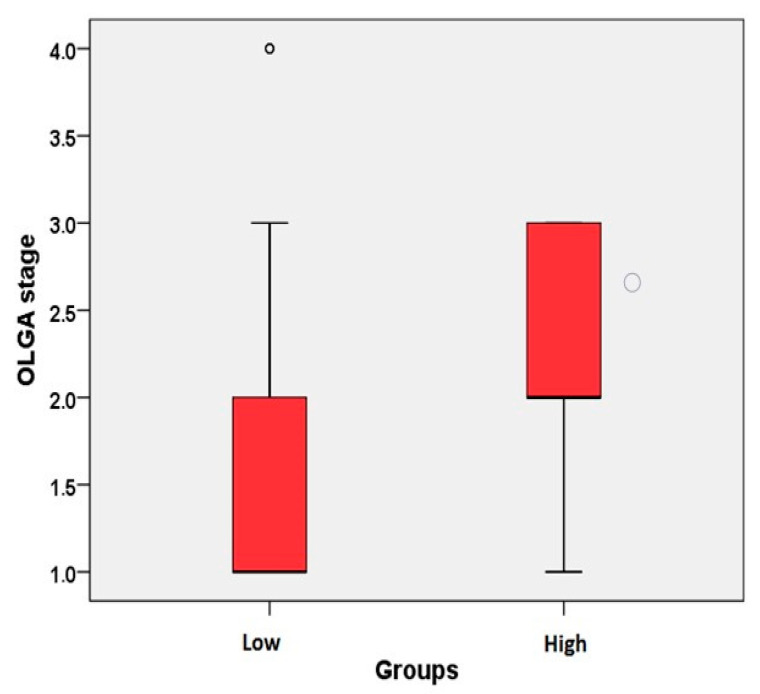
Comparison of the OLGA stage according to the neutrophil to lymphocyte ratio (NLR) and the NRS groups (*n* = 14, *n* = 141).

**Table 1 healthcare-08-00230-t001:** Correlation analysis results between NRS and gastric parameters (*n* = 155).

		A Score	C Score	OLGA Stage	Intestinal Metaplasia	Inflammation	Activity
NRS	R	0.377 *	0.396 *	0.469 *	0.223 *	0.072	0.080
	*p*	<0.001	<0.001	<0.001	0.005	0.374	0.320

* Spearman’s rho correlation coefficient statistically significant, NRS: Numerical Rating Scale.

**Table 2 healthcare-08-00230-t002:** Correlation analysis results between NLR and gastric parameters (*n* = 155).

		A Score	C Score	OLGA Stage	Intestinal Metaplasia	Inflammation	Activity
NLR	R	0.106	0.258 *	0.208 *	0.080	0.118	0.096
	*p*	0.191	0.001	0.009	0.325	0.145	0.234

* Spearman’s rho correlation coefficient statistically significant, NLR: neutrophil/lymphocyte ratio.

**Table 3 healthcare-08-00230-t003:** Comparison of HP incidence according to neutrophil to the lymphocyte ratio (NLR) and numerical rating scale (NRS) levels.

		HP		Total	*p* Values
		−	+		
NLR	Normal (≤3.5)	72	62	134	0.354
		53.7%	46.3%	100%	
	High (>3.5)	9	12	21	
		42.9%	57.1%	100%	
NRS	Normal (<4)	21	27	48	0.156
		43.8%	56.2%	100%	
	High (≥4)	60	47	107	
		56.1%	43.9%	100%	
NRS and NLR	Negative	76	65	141	0.194
		53.9%	46.1%	100%	
	Positive	5	9	14	
		35.7%	64.3%	100%	
Total		81	74	155	
		52.3%	47.7%	100%	

Chi square test.

**Table 4 healthcare-08-00230-t004:** Comparison of gastric parameters according to NLR and NRS groups (*n* = 155).

	NLR and NRS	*N*	Mean ± SD	Median (min–max)	*p* Values
A score	−	141	1.43 ± 0.55	1 (0–3)	0.131
	+	14	1.64 ± 0.49	2 (1–2)	
C score	−	141	1.07 ± 0.48	1 (0–3)	0.001 *
	+	14	1.5 ± 0.51	1.5 (1–2)	
OLGA stage	−	141	1.59 ± 0.73	1 (1–4)	0.007 *
	+	14	2.14 ± 0.77	2 (1–3)	
Intestinal metaplasia	−	141	0.25 ± 0.52	0 (0–3)	0.578
	+	14	0.29 ± 0.46	0 (0–1)	
Inflammation	−	141	2.21 ± 0.63	2 (1–3)	0.106
	+	14	2.50 ± 0.51	2.5 (2–3)	
Activity	−	141	1.23 ± 1.01	1 (0–3)	0.244
	+	14	1.57 ± 1.08	2 (0–3)	

* Mann Whitney U test statistically significant, NRS: Numerical Rating Scale, NLR: neutrophil/lymphocyte ratio, SD: standard deviation.

**Table 5 healthcare-08-00230-t005:** Demographic and clinical descriptive statistics of patients.

		*n* (%)/Mean ± SD (min–max)
Gender		
	Female	84 (54.2%)
	Male	71 (45.8%)
OLGA		
	1	79 (51%)
	2	56 (36.1%)
	3	17 (11%)
	4	3 (1.9%)
Age		49.1 ± 15.2 (18–81)

SD: Standard deviation.

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
