# Peer review of "Can Simple Tests Prior to Endoscopy Predict the OLGA Stage of Gastritis?"

_healthcare, 2020, doi:10.3390/healthcare8030230_

Round 1

Reviewer 1 Report

Authors would associate the severity of gastritis  with a simple complete blood count. However, the correlation of neutrophil to lymphocyte ratio for gastric atrophy is not new and did not improve significantly Olga stage stratification. Results and discussion of the paper does not add important information regarding the stratification of atrophy.

Reviewer 2 Report

General: The authors have identified an interesting topic here on “Can simple tests prior to endoscopy predict gastric atrophy? An association between pain, neutrophil to lymphocyte ratio and OLGA stage for gastritis” which is an interesting topic.

Some concerns which will need to be addressed are:

Title: looks too long, would advise the authors to shorten it

Abstract:

Looks ok

Key words:

Please use full forms rather than abbreviation in key words

Introduction:

Looks ok

Subjects and methods:

Line 77: authors mention “to divers reasons”…please explain or correct it

Intensity of pain is subjective: one patient might complain of pain 2/10 and the same pain rating for another patient might be 7/10. How did authors exclude this bias

Please discuss about any confounders to the study and how they were excluded

Results:

Look ok

Discussion:

Recent study has shown that “between NLR and platelet to lymphocyte ratio might an even better predictor complications than NLR alone” please add this to the current discussion. https://journals.lww.com/md-journal/FullText/2019/06280/The_usefulness_of_inflammatory_biomarkers_in.67.aspx

Limitations should be discussed in detail.

Tables are appropriate.

English and grammar needs to be thoroughly rechecked.

Overall a well conducted study.

Reviewer 3 Report

Introduction:

Line 52: '... damage to ...'

Line 53: '... most cases ...'

Line 57: '... gastritis may ...'

Line 58: '... visible vessels ...'

Line 60: Need to define OLGA and OLGIM in the Introduction since it is separate from the Abstract.

Line 63: '... are directly ...'

Line 68: 'has been' instead of 'was'

Line 70: '... of illness severity ... use in a wide spectrum of diseases, namely ...'

Line 71: '... such as in cancer treatment or coronary ...'

Methods:

Line 76: '... upper endoscopy. Out of ...'

Line 77: '... to several reasons, namely inadequate sampling or intolerance  ...'

The exclusion criteria should be described first. Then the 45 patients who were lost to follow-up should be described. Figure 1's formatting should be improved to better size the boxes, align the arrows, and streamline the text.

Line 122: 'The staff consisted of ...'

Line 125: '... a senior staff member (IO).'

Line 129: Add a space between 'in' and '10%'

Line 134: 'All patients undergoing ...'

Lines 151 and 153 seem duplicated aside from the effect size; which was the one used in the study?

Results:

Line 187: A moderate correlation would usually be considered with an r value of 0.5 to 0.7; I think this is close but would likely be considered weak-moderate by many statisticians.

Figure 3: x-axis label should be 'positive' instead of 'pozitive' Also, it would be good to emphasize that very few patients had both high NLR and high NSR in the text to support the rationale for this figure. I would suggest calling it NSR/NLR High versus NSR/NLR Low instead of positive/negative.

The Discussion contains a great deal of information, some of which should be moved to the Introduction to better support the study rationale.

Overall, the identified correlations should be better explained to highlight to the reader that these are weak correlations, otherwise they are being oversold. 

Round 2

Reviewer 1 Report

How you evaluate in detail the scale for NRS??, possibly add the questionnaire for readers who are not accustomed to this wording.

The authors must also better detail the case studies, for example, how many cases are olga 3 and how many olga 4 ??  There is no reported the descriptive table of demographic/clinical patient series.

The choice of some statistical tests is not comprehensible if they are not justified, e.g. why a regression with a quadratic curve is made and not a linear regression ?? NLR 0.1 effect size, NLR 0.2 effect size ?? While the difference in the size of the effect ??

There is a correlation between NLR and gastric atrophy, while no correlation was found for metaplasia. This is strange and suggests that the association is not accidental but only occasionally obtained or due to a low number of cases with olga 3 and olga 4 with metaplasia.  Any correlation between the intensity of gastric pain and H. Pylori is rare, also this affermation is strange and not justified.

Minor points

minimal typographical mistakes

lane 45 CBC …the name for the abbreviation  is absent, as well as  lane 105 staff IO ???

lane 139: file number is not a Demographic data and patient medical history

lane 142 : NRS then NRS end pts ??? please, specify

lane 144: frequency of administration….administration of WHAT ?

lane 232: the sentence “Recently it has proven to be a useful marker in predicting, assessing, and stratification of several diseases and conditions such as inflammatory diseases, benign or malignant maladies, prediction of postoperative mortality and complications”  ..have no references. Please include at least one to support this hypothesis.

Reviewer 3 Report

This revision is much improved and more appropriately states the results and conclusions of the study. Some text in the Discussion session would benefit from review by a native English speaker for word choice and flow.

Author Response

I would like to thank reviewer-3- for the valuable comments and criticisms that had made to enrich our paper.

Comment from Reviewer-3: Some text in the Discussion session would benefit from review by a native English speaker for word choice and flow

Author response: As requested by the reviewer our paper was revised by an English language editing service and herein we have attached the English language Editing Certificate.
